# *Bna.EPF2* Enhances Drought Tolerance by Regulating Stomatal Development and Stomatal Size in *Brassica napus*

**DOI:** 10.3390/ijms24098007

**Published:** 2023-04-28

**Authors:** Peipei Jiao, Yuanlin Liang, Shaoping Chen, Yang Yuan, Yongqiang Chen, Honghong Hu

**Affiliations:** 1Key Laboratory of Crop Genetic Improvement, Hubei Hongshan Laboratory, Huazhong Agricultural University, Wuhan 430070, China; jiaopeipei2000@126.com (P.J.); yuanlinliang2019@126.com (Y.L.); spchen0929@163.com (S.C.); yangyuan9109@163.com (Y.Y.); chenyq2017@163.com (Y.C.); 2Xinjiang Production and Construction Corps Key Laboratory of Protection and Utilization of Biological Resources in Tarim Basin, College of Life Science, Tarim University, Alar 843300, China

**Keywords:** *Bna.EPF2*, *Brassica napus*, drought tolerance, stomatal density, stomatal dimension, water use efficiency

## Abstract

Drought stress severely affects global plant growth and production. The enhancement of plant water-use efficiency (WUE) and drought tolerance by the manipulation of the stomata is an effective strategy to deal with water shortage. However, increasing the WUE and drought tolerance by manipulation on the stomata has rarely been tested in *Brassica napus*. Here, we isolated *Bna.EPF2*, an epidermal patterning factor (*EPF*) in *Brassica napus* (ecotype Westar), and identified its role in drought performance. *Bna.EPF2* overexpression lines had a reduction average of 19.02% in abaxial stomatal density and smaller stomatal pore size, leading to approximately 25% lower transpiration, which finally resulted in greater instantaneous WUE and enhanced drought tolerance. Interestingly, the reduction in stomatal density did not affect the CO_2_ assimilation or yield-related agronomic traits in *Bna.EPF2* overexpression plants. Together with the complementation of *Bna.EPF2* significantly decreasing the stomatal density of Arabidopsis *epf2*, and *Bna.EPF2* being expressed in mature guard cells, these results suggest that *Bna.EPF2* not only functions in stomatal density development, but also in stomatal dimension in Brassicas. Taken together, our results suggest that *Bna.EPF2* improves WUE and drought tolerance by the regulation of stomatal density and stomatal size in Brassica without growth and yield penalty, and provide insight into the manipulation of this gene in the breeding of drought tolerant plants with increased production under water deficit conditions.

## 1. Introduction

Drought stress is one of the most destructive environmental stresses to plants. It severely affects plant development and growth, and becomes a key factor limiting plant productivity [1,2]. The stomata function as key valves that regulate the gas exchange between plants and the environment, by uptaking CO_2_ for photosynthesis and losing water for transpiration. Reducing transpiration rates via the stomata is efficient for most plants under water deficit, and they thus increase their water use efficiency (WUE) [3]. Therefore, reducing transpiration and enhancing the WUE of plants under drought stresses without yield penalties has become the main objective for plant breeding programs. 

Plants can regulate transpiration and WUE by managing stomatal conductance, which is mainly determined by stomatal density and stomatal movement [4,5,6,7,8,9]. For short-term water deficit, plants close the stomata to reduce transpiration and improve WUE. Under long-term water deficit conditions, plants produce maximum stomatal conductance by changing the stomatal size and density [10,11]. Therefore, manipulating the stomata number is one of the ways to alter the photosynthetic efficiency and water transpiration of plants [12,13,14], thus improving plant physiological characters and providing the potential for increasing crop yield.

In dicots, stomatal development begins with an asymmetric division of the meristematic mother cell (MMC), which generates a small triangular-shaped meristemoid (M) and a large stomatal lineage ground cell (SLGC). The SLGC differentiates into an epidermal pavement cell or another meristemoid, whereas an M then differentiates into a guard mother cell (GMC). Finally, the GMC divides symmetrically to form a pair of guard cells (GC) [14]. EPFs polypeptide hormones, encoded by EPIDERMAL PATTERNING FACTOR family members, are secreted as an extracellular signal for stomatal development initiation, and the receptors TMM (too many mouths) and ERECTA in the surrounding cells perceive these extracellular signals [15,16,17], and then transfer the signals across the membrane to intracellular components by the MAPK signaling pathway to initiate or inhibit stomatal development [18,19,20,21]. There are a total of 11 members of the EPF family in Arabidopsis, including EPF1, EPF2, and nine EPFLs [20,21,22,23]. Four members of the EPF family, EPF1, EPF2, EPFL9, and EPFL6, were found to participate in the process of stomatal development [23,24,25,26,27,28]. EPF1, EPF2, and EPFL6 are negative regulators, and EPFL9 is a positive regulator of stomatal density by primarily controlling the initiation and frequency of asymmetric cell divisions and finally determining the occurrence and density of stomata in Arabidopsis [20,23,26,29]. EPFL9, also named STOMAGEN, regulates stomatal development by antagonizing EPF1 and EPF2 [27,28,30]. Due to the roles of EPFs in stomatal development, their effects on water relations and net photosynthesis have been tested in Arabidopsis, crops, and trees. The overexpression of *EPF2* significantly improved drought tolerance without deleterious effects on the nutrient uptake in Arabidopsis [12,31]. The ectopic overexpression of *PdEPF2* enhanced drought tolerance by modulating stomatal density and regulated the ABA response of *Arabidopsis* transgenic plants [32], and the ectopic overexpression of *MdEPF2* from apple improved WUE and reduced oxidative stress in tomato [33]. The overexpression of *EPF1* homologs improved plant WUE and drought tolerance by reducing stomatal density in poplar and wheat [34,35]. The overexpression of *OsEPF1* in rice plants showed reduced stomatal density and increased root cortical aerenchyma formation [36], and *HvEPF1* overexpression in barley caused significantly enhanced WUE and drought tolerance [29]. The genetic engineering of *EPFL9/Stomagen* or the spraying of stomagen peptide led to high stomatal densities and crop production, in which the photosynthetic rate was enhanced by an average of 30% [12,28,37]. EPFL family genes exist in many land plants. However, no such reports have been reported in Brassica*,* and whether the functions of the EPFL family in the dicotyledon Brassica are also conserved remains unclear.

*Brassica napus* is a major oilseed crop in China and approximate 20% of the world’s yield is produced by China [38]. However, the high productivity and quality strongly depends upon water availability [39], and the trade-offs between growth and stress resistance in plants has long been a research focus for scientists and research. Although most previous studies of *EPF2* on both herbaceous and woody plants suggested vital roles in regulating stomatal development, whether it performs the same function in *Brassica napus* needs further experimental evidence. In this study, we focused on the homologous gene of *EPF2* in *Brassica napus* and identified its role in stomata development and the effect on water use efficiency, drought tolerance, and yield traits. The expression of *Bna.EPF2* clearly decreased stomatal density in *Brassica napus* and restored the stomatal phenotype of *Arabidopsis epf2* mutant plants. The *Bna.EPF2* overexpression plants had higher WUE, lower transpiration rate and stomatal conductance at normal growth conditions and increased drought tolerance compared to wild-type plants under water limited conditions. Interestingly, the seed yield per plant was not affected in these overexpression plants. This study revealed the conserved role of Bna.EPF2 in stomatal development, and a new role in stomatal dimension compared to other EPF2 orthologs was reported which coordinated with the water use efficiency of plants. Our results suggested the potential of Bna.EPF2 in crop drought tolerance improvement and seed yield per plant and provided a new gene resource for genetic manipulation.

## 2. Results 

### 2.1. Isolation and Characterization of Bna.EPF2 in Brassica napus

To identify orthologs of EPF and EPFL family members in *Brassica napus*, all 11 previously reported EPF and EPFL peptides in Arabidopsis were used as a BLASTP query to search the *Brassica napus* genome (http://plants.ensembl.org/Brassica_napus, AST_PRJEB5043_v1, accessed on 1 September 2014). A total of 25 putative EPF/EPFLs were obtained. A phylogenetic tree of these predicted Bna.EPFL peptides with those from Arabidopsis were constructed using MEGA 6.06 and the maximum likelihood method (Figure 1A). Each Arabidopsis EPF/EPFL had 1–2 orthologs in *B. napus*, in A and/or C genomes. *EPF1*, *EPF2*, *EPFL9*, and *EPFL6* were reported to be involved in stomatal development in Arabidopsis [20,23,26]. As shown in Figure 1A, there were two closed Arabidopsis *EPF1* orthologs in *B. napus*, one from the A genome (BnaA09g43550D) and the other from the C genome (BnaC08g36150D). *EPFL6* and *EPFL9* also had two orthologs in *B. napus*, respectively, with one each in the A and C genomes (Figure 1A), whereas only one closed ortholog of *EPF2*, *BnaA09g23140D*, was presented in the *B. napus* genome and, therefore, it was named *Bna.EPF2* in this study. Two orthologs of EPF1, EPFL6, and EFPL9 in Brassica showed high sequence similarity (Appendix A).

Bna.EPF2 had 119 amino acids (AAs), including a 25AA long signal peptide, and a 49AA EPF domain (https://smart.embl-heidelberg.de, 3 March 2023, version 9), with 13.17 KD molecular weight and 9.10 isoelectric point (https://web.expasy.org/cgi-bin/protparam/protparam, 3 March 2023). Phylogenetic analysis and the multiple sequence alignment of EPF2s in both dicots (*Brassica napus*, *Arabidopsis thaliana*, and *Glycine max*) and monocots (*Sorghum bicolor*, *Zea mays*, and *Oryza sativa*) indicated that Bna.EPF2 was the most closed ortholog of AtEPF2, with an 81.67% protein identity (Figure 1B). These results indicated that Bna.EPF2 may have a similar function in stomatal development to AtEPF2. EPFs have a signal peptide at the N-terminus, which needs to be cleaved before they function. Through a comparison of the sequence of Bna.EPF2 with other orthologs and the prediction of the cleavage site as reported in AtEPF2 [40], the mature peptide of Bna.EPF2 was found to be 45AA in length, and there was only 1 amino acid different from AtEPF2 (Figure 1B). Therefore, we chose *Bna.EPF2* for further functional identification.

### 2.2. Expression Pattern of Bna.EPF2

To examine the expression pattern of *Bna.EPF2*, we generated transgenic lines expressing the β-glucuronidase (GUS) reporter gene under the control of the promoter of the *Bna.EPF2* gene (Figure 2A). Histochemical GUS staining showed a high expression of *Bna.EPF2* in young leaves, but not in roots or stems at the seedling stage. *Bna.EPF2* was also expressed in flowers and siliques at the reproductive stage, with a higher expression in calyx, ovary, and stigma, but not in petals and stamens (Figure 2B). At the cellular level, *Bna.EPF2* likely had a preferential expression in the stomatal lineage cells of leaves and calyx, with a great expression in mature guard cells (Figure 2C). In contrast to Arabidopsis *EPF2*, *Bna.EPF2* was only expressed in stomatal precursor cells, and enriched in the early meristemoid mother cell (MMC) [23]. These results revealed that *Bna.EPF2* had a similar but expanded expression pattern compared to the *AtEPF2* gene in Arabidopsis, indicating that Bna.EPF2 may have a broader function than AtEPF2.

### 2.3. The Ectopic Expression of Bna.EPF2 Rescued the Stomatal Development Defects in Arabidopsis epf2 Mutant

The mutation of Arabidopsis *EPF2* caused increased stomatal density, and its overexpression plants showed a phenotype of reduced stomatal density [12]. To explore whether *Bna.EPF2* is also involved in stomatal development and has a similar function to AtEFP2, we expressed *Bna.EPF2* CDS driven by 35S promoter in the Arabidopsis *epf2* mutant plants, in which the stomatal density was increased as reported (Figure 3A) [26]. We found that the expression of *Bna.EPF2* dramatically reduced the stomatal density of *epf2* (Figure 3B). Statistical analyses showed that these three randomly selected independent lines had an even lower level of stomatal density than Col-0, with an average of 13.04–20.49% decline compared to Col-0 (Figure 3A,B). Stomatal index, the ratio of stomata to all epidermal cells (pavement cells and stomata) in a given area (per mm^2^), a more accurate indicator reflecting stomatal development, was also decreased by 14.04–22.91% in these *Bna.EPF2* expression *epf2* transgenic lines (Figure 3C). These data suggested that Bna.EPF2 has a conserved function as AtEPF2, negatively regulating stomatal development.

### 2.4. Overexpression of Bna.EPF2 in Brassica Plants Caused a Lower Stomatal Density and Smaller Stomatal Complex Size

To further investigate the function of *Bna.EPF2*, we overexpressed *Bna.EPF2* into the *B. napus* species Westar. Three transgenic lines, *OE-17*, *OE-19*, and *OE-29*, in which *Bna.EPF2* was overexpressed, were randomly selected for the determination of stomatal density and index phenotypes (Figure 4A). *Bna.EPF2* overexpression plants exhibited an average of 19.02% lower stomatal density and 6.90% lower stomatal index than WT in true leaves (Figure 4B–D), as well as an average of 28.96% lower stomatal density and 9.05% lower stomatal index than WT in cotyledons (Appendix A), consistent with the role of EPF2 in Arabidopsis. These results further suggested that Bna.EPF2 and AtEPF2 maintained functional conservation in stomatal development. 

In addition to the stomatal density changes, we found, interestingly, that the stomatal complex size was also altered in the *Bna.EPF2* overexpression plants (Figure 4E–J). *Bna.EPF2* overexpression plants had a smaller stomatal complex size, with reduced complex length and/or width (Figure 4F,G). Stomatal pore length, pore width, and pore area were also reduced by an average of 6.78%, 4.80% and 10.90% in these transgenic lines (Figure 4H–J), respectively. Together with the fact that *Bna.EPF2* was expressed in young and mature guard cells (Figure 2C), these results suggested that *Bna.EPF2* is also involved in stomatal dimension regulation, in addition to its regulation in stomatal density development.

### 2.5. Overexpression of Bna.EPF2 in B. napus-Conferred Enhanced Drought Tolerance

As the stomata control CO_2_ influx and the loss of transpiration water, changes in plant transpiration caused by stomatal density and dimension changes will affect the leaf temperature and drought performance. We therefore analyzed the leaf temperature of *Bna.EPF2* overexpression lines and WT plants at normal growth conditions. The leaf temperature was 0.23 °C–1 °C higher in the *Bna.EPF2* overexpression plants than that of WT plants (Figure 5A,B), consistent with the reduced stomatal density and stomatal pore area in the *Bna.EPF2* overexpression plants (Figure 4B,J). Next, drought stresses were applied to 14-day-old WT and *Bna.EPF2* overexpression plants. During drought stresses, the wild-type plants wilted earlier than the *Bna.EPF2* overexpression plants (Figure 5C,D). When water was resupplied for five days, an average of 70.14% of the *Bna.EPF2* overexpression plants were recovered, whereas only about 8.33% of the WT plants were recovered (Figure 5C,D), suggesting that *Bna.EPF2* confers drought tolerance in Brassicas.

### 2.6. Overexpression of Bna.EPF2 in B. napus-Improved WUE by Reducing Transpiration 

To explore the impact of decreased stomatal density and stomatal pore size in the *Bna.EPF2* overexpression plants, stomatal conductance, leaf transpiration, and instantaneous WUE (photosynthesis/transpiration) were analyzed using a Li-6400XT Photosynthesis system. The stomatal conductance at the steady state under 450 ppm CO_2_ conditions was reduced by 19.65–33.04% in these transgenic *Brassica* plants (Figure 6A), which was mainly caused by their lower stomatal density and stomatal pore area (Figure 4B,J). Consistent with the reduced stomatal conductance, the *Bna.EPF2* overexpression plants had a decrease of 21.57–31.59% in the leaf transpiration rate (Tr) compared to WT plants (Figure 6B), and a significant 3.14–5.49% higher instantaneous WUE (iWUE) (Figure 6C), which may explain the enhanced drought tolerance of the *Bna.EPF2* overexpression plants under drought stress conditions. 

To further determine the effects of stomatal density on leaf gas exchange, CO_2_ assimilation, leaf transpiration, and instantaneous WUE under a range of external CO_2_ concentrations and light intensities were also examined. A-Ci curves and light curves showed that the *Bna.EPF2* overexpression plants had a similar level of assimilation rate under a range of CO_2_ concentrations and saturating light levels (Figure 6D,G). The transpiration of *Bna.EPF2* overexpression plants was consistently reduced under different CO_2_ concentrations and light intensities, with a 17.64–26.5% reduction under different CO_2_ conditions and a 13.96–37.54% reduction under different light intensities compared to WT (Figure 6E,H), matching well with their reduced stomatal density and dimension (Figure 4). The *Bna.EPF2* overexpression lines had a comparable or even higher instantaneous WUE than WT in response to the shifts in light intensity or CO_2_ concentrations (Figure 6F,I). These results demonstrated that the overexpression of *Bna.EPF2* affected leaf transpiration by regulating stomatal density and dimension in *B. napus*, further suggesting the potential role of *Bna.EPF2* in the improvement of WUE and drought tolerance in plants. 

### 2.7. Overexpression of Bna.EPF2 Did Not Influence Major Agronomic Traits 

The decreased stomatal density and smaller stomatal pores in the *Bna.EPF2* overexpression plants may have affected plant growth and yield under the natural field. To explore this, we measured the agronomic traits of the transgenic Brassica plants that grew in a Wuhan natural field. *Bna.EPF2* overexpression had no significant impact on plant height, the length of main inflorescence, the number of siliques on main inflorescence, the number of the first branch, 1000-seed weight, or seed yield per plant, which were all similar to those of WT plants. Interestingly, the length of the silique and number of seeds per silique were significantly increased, with an average increase of 4.46% and 4.02% in the *Bna.EPF2* transgenic lines, respectively (Table 1), consistent with the continuous expression of *Bna.EPF2* in the ovary and young siliques of transgenic plants (Figure 2B). These results indicated that the overexpression of *Bna.EPF2* reduced stomatal density and dimension without yield penalty.

### 2.8. Bna.EPF2 Affects Stomatal Density via Different Stomatal Development Pathways in B. napus

*Bna.EPF2* and *AtEPF2* have similar functions in stomatal development, indicating that there is a common pathway for stomatal development in both *Arabidopsis* and *B. napus*. Three bHLH transcription factors, SPCH, MUTE, and FAMA, are key regulators of stomatal development, from the initial transition of a protodermal cell to mature guard cells in Arabidopsis, and may be conserved from basal land plants through to angiosperms. In addition, the subtilisin-like serine protease *STOMATAL DENSITY AND DISTRIBUTION1* (*SDD1*) acts as a negative regulator of stomatal development independent of the main signaling pathway. We searched the *SPCH*, *MUTE*, *FAMA*, and *SDD1* homologous genes in *B. napus* using BLAST alignments (https://plants.ensembl.org/Brassica_napus, accessed on 3 March 2023), and found that each had multiple copies in *B. napus* (Appendix A). Therefore, we selected *BnaC02G14500D*, *BnaA03g57400D*, *BnaA03g37180D*, and *BnaA10g02390D* as the candidate genes for *Bna.SPCH*, *Bna.MUTE*, *Bna.FAMA*, and *Bna.SDD1*, respectively, and determined their expression levels in the transgenic plants, since they had the highest protein identities compared with Arabidopsis. The relative expression levels of *Bna.SPCH* and *Bna.MUTE* were significantly decreased in *Bna.EPF2* overexpression lines compared to WT, and *Bna.FAMA* was not significantly altered (Appendix A), consistent with the phenotypes of decreased stomatal density in the *Bna.EPF2* overexpression lines (Figure 4B), as well as in the Arabidopsis *spch* and *mute* mutant plants [40]. Surprisingly, the expression level of *Bna.SDD1* was also significantly decreased. These results suggested that *Bna.SPCH* and *Bna.MUTE* have similar roles in stomatal development to Arabidopsis, and the overexpression of *Bna.EPF2* may affect the stomatal density via the transcriptional inhibition of *SPCH* and *MUTE* in different ways, thereby inhibiting the asymmetric cell division and thus regulating stomatal density.

## 3. Discussion

Drought stress is one of the major adverse environmental stresses, causing damage to crop development and huge yield loss. As one of the most important oil crops, *B. napus* ranks first in the proportion of sowing area and total yield in the world. In the context of climate change, environmental stresses and their impact on yield formation are becoming more and more prominent. It is a big challenge in crop breeding to identify genes that confer drought tolerance, but have no significant penalty on yield. The stomata function as valves for gas exchange, therefore, manipulating the stomatal density is a good way for breeding plants to cope with water shortage conditions.

Stomatal development has been well studied in Arabidopsis, with many genes having been reported to participate in this process. *EPFs/EPFLs* are the most important ones that play key roles in stomatal development and distribution [41], and they may represent a potential target for crop improvement [29,31,32,33,34,42,43]. Sequence and phylogenetic analyses showed that there was only one closed *EPF2* ortholog in the genome of *B. napus*, named *Bna.EPF2* in this study (Figure 1). Based on the findings that *Bna.EPF2* expression rescued the increased stomatal density phenotype of Arabidopsis *epf2* (Figure 3B), and that *Bna.EPF2* overexpression caused decreased stomatal density and index in Brassica Westar (Figure 4B,C), these results suggested that *Bna.EPF2* was also involved in stomatal development, with a similar function to Arabidopsis *EPF2*. Changing the expression of *EPF2* orthologs in other plants, such as barley, poplar, and apple, also influenced the stomatal development [29,32,33], suggesting that the function of EPF2s in stomatal density is highly conserved in plants. As *SPCH* and *MUTE* regulate distinct stomatal developmental steps [44,45,46,47,48], *Bna.SPCH* and *Bna.MUTE* were significantly reduced in *Bna.EPF2* overexpression Brassica plants (Appendix A), which coincided with results in Arabidopsis, suggesting that EPF2 peptides control asymmetric cell division in the early stomatal lineage and restrict entry into the stomatal lineage or promote an exit to the pavement cell fate, and thus regulate stomatal density. In contrast to *AtEPF2* [26], *Bna.EPF2* was also enriched in young and mature guard cells (Figure 2C) and its overexpression in Brassica caused a smaller stomatal complex size and stomatal pore size (Figure 4). These results suggested that *Bna.EPF2* has a broader function in stomatal biology. In addition to its conserved role in stomatal density regulation, it also plays a role in stomatal dimension. The function of EPFs in stomatal dimension has not yet been reported, and therefore the ways in which *Bna.EPF2* regulates stomatal dimension can be investigated in the future, which may lead to the exploration of new roles of EPFs in stomatal biology.

The iWUE and drought tolerance were significantly enhanced in *Bna.EPF2* overexpression transgenic plants (Figure 5C and Figure 6C), in line with a reduction in the stomatal density and stomatal pore area (Figure 4B,J). These results suggested that the manipulation of stomatal development in Brassica improves WUE and drought tolerance. The gas exchange is controlled by stomata and influenced by stomatal movements (opening and closing), stomatal density, and pore size [49,50,51,52]. The CO_2_ assimilation rate, stomatal conductance, and transpiration rate of plants change significantly under abiotic stresses [53,54], indicating that the reduction in stomatal conductance may affect CO_2_ assimilation rates and limit the production of biomass due to the limitations of gas exchange [55]. However, recent reports have showed that a small reduction in the stomatal density did not has a significant impact on the biomass or yield. For example, Arabidopsis *EPF2* overexpression plants that had an 80% reduction in stomatal density showed only a small reduction in net photosynthesis at high light, *EPFL9-RNAi* plants with a 68% reduction in stomatal density showed no significant differences in photosynthesis compared to Col-0 plants [42,56], and *HvEPF1* overexpression barley plants had a 58% reduction in stomatal density and showed no reductions in seed yield [29]. Additionally, the overexpression of both *HARDY* and *HDG11*, and mutations of *GTL1* and *GPA1*, reduced stomatal density and improved drought tolerance with no penalty on carbon assimilation or biomass accumulation [50,57,58,59]. In our study, *Bna.EPF2* overexpression Brassica transgenic plants showed a lower stomatal conductance and a comparable CO_2_ assimilation rate under various CO_2_ concentrations and light intensities (Figure 6), which thus resulted in no penalty on the plant growth and seed yield related agronomic traits (Table 1). A possible explanation is that a suitable decrease in the total stomatal pore area (stomatal density and single stomatal pore area) in leaves might have no impact on the total amount of CO_2_ necessary for photosynthesis, and the CO_2_ supply in these lines was still enough to drive photosynthesis for normal growth under natural field conditions. 

In conclusion, we identified the role of the EFP2 ortholog Bna.EPF2 in Brassica, which had a conserved role in stomatal density development and a new, unreported role in stomatal dimension. The overexpression of *Bna.EPF2* conferred an enhanced drought tolerance on *B. napus* plants without a penalty on plant growth or yield. Our results provided an insight into the mechanism of *Bna.EPF2* in the regulation of WUE and drought tolerance, and suggested the potential for the manipulation of one gene in the improvement of crop drought tolerance by the alteration of stomatal density and pore size.

## 4. Materials and Methods

### 4.1. Plant Materials and Growth Conditions

*Brassica napus* (ecotype Westar) and *Arabidopsis thaliana* (ecotype Col-0) were used in this study. The single T-DNA insertion mutant *epf2* (SALK_047918) in Arabidopsis was obtained from the Arabidopsis Biological Resource Center (ABRC).

Arabidopsis and Brassica seeds were surface-sterilized with 75% ethanol for 5 min and 95% ethanol for 30 s, and dried. The seeds were germinated for 7d in a 1/2 MS medium plate at 21 °C after being vernalized at 4 °C for 3 d. The Arabidopsis and Brassica plants were transferred to a compost soil mix (peat moss, perlite, and vermiculite, 1:1:1, *v*/*v*/*v*) and grown in a well-controlled growth chamber or a greenhouse at 22 °C with a 16 h light/8 h dark photoperiod regime.

### 4.2. Sequence Retrieval and Phylogenetic Tree Construction

The amino acid sequences of putative EPFs and EPFLs were obtained from the Arabidopsis Information Resource (TAIR) database (http://www.Arabidopsis.org, 3 March 2023), the *Brassica napus* genome database (https://www.genoscope.cns.fr/brassicanapus, http://plants.ensembl.org, accessed on 3 March 2023), and the National Center for Biotechnology Information (NCBI) database. The sequences were aligned using MEGA 6.06 software, and a phylogenetic tree was constructed using maximum likelihood method (bootstrap 1000 replications) with default parameters. 

### 4.3. Generation of pBna.EPF2::GUS Transgenic Lines and GUS Staining

To generate the *pBna.EPF2*::*GUS* expression vector, a 2 kb promoter region upstream of the ATG start codon was amplified by PCR with a primer pair *Bna.EPF2-pF* and *Bna.EPF2-pR* using wild type *B.napus* Westar genomic DNA as a template. The fragment digested by *Hind*III and *Bam*HI was subcloned into the corresponding restriction sites on the expression vector *pLP100* containing a *GUS* reporter gene [60,61]. The promoter sequence was confirmed by DNA sequencing, and the primers used were listed in Appendix A. The *pBna.EPF2*::*GUS* construct was transformed into Col-0 using the floral dipping method [62], and transgenic plants were screened on 1/2 MS agar medium containing 50 μg·mL^−1^ of Kanamycin; the positive transgenic plants were further confirmed by detecting the existence of the GUS reporter gene. The representative T1 lines showing consistent GUS staining were further analyzed for GUS staining at the seedling stage and reproductive stage. At least three independent transgenic lines were analyzed in parallel. For GUS staining, seedlings, flowers and siliques of *pBna.EPF2*::*GUS* transgenic Arabidopsis plants were incubated in GUS staining buffer overnight at 37 °C, following the procedure as described [63]. For the observation of expression in stomatal lineage cells, the abaxial leaf epidermal layers, in which mesophyll cells were not in the vicinity, were stained in GUS staining buffer at 37 °C for 2 h. After staining, the seedlings and all tissues were de-colored or rinsed with 75% ethanol several times for GUS observation under a microscope.

### 4.4. Generation of Transgenic Arabidopsis and Brassica napus Plants 

The coding sequences (CDSs) of *Bna.EPF2* (BnaA09g23140D, *Brassica napus* AST_PRJEB5043_v1) were amplified from cDNA of *Brassica* leaves with primers *Bna.EPF2*-F and *Bna.EPF2*-R. The PCR products were cloned into the 35S promoter-driving pBI121 binary vector by double digestion with *SmalI* and *XbaI.* The *35S*::*Bna.EPF2* construct was transformed into the *Agrobacterium* strain GV3101(pMP90) and then into Arabidopsis plants using the floral dipping method [62], and into *Brassica napus* by Agrobacterium-mediated procedures as described [64]. Transgenic plants were initially selected on 1/2 MS agar medium containing 50 μg·mL^−1^ of Kanamycin, and further confirmed by the determination of select-marker gene *NPTII* with PCR. The expression level of positive *Bna.EPF2* overexpression lines was analyzed by qPCR analyses, with gene-specific primers listed in Appendix A. 

### 4.5. Thermal Imaging Analysis 

Thermal imaging was performed on 3-week-old *Brassica napus* plants under normal growth conditions using an infrared camera (FLIR systems; T420), as described [65]. Leaf temperatures were measured in fully expanded first and second true leaves using FLIR Tools^+^ ver. 5.2 software. 

### 4.6. Transpiration and WUE Measurements

Photosynthetic and transpiration rates were measured in fully expanded second true leaves of 5-week-old *Brassica napus* plants using a Li-6400XT Photosynthesis system (LI-6400 Inc., Lincoln, NE, USA). The measurements were applied under the condition of a photon flux density of 300 μmolm^−2^ s^−1^, cuvette flow of 300 mL min^−1^, and an ambient CO_2_ concentration of 450 ± 5 μmolmol^−1^ at 21 °C. The rates were recorded every 30 s for 10 min. Rates of photosynthesis (Pn), transpiration (Tr), and stomatal conductance (Gs) were obtained from at least three plants for each genotype per experiment. Instantaneous water use efficiency (iWUE), defined as the ratio of CO_2_ assimilated to water lost during transpiration (μmol CO_2_ mmol H_2_O^–1^), was calculated from the data of transpiration and photosynthetic rate. Three independent experiments were performed. 

A-Ci curves were determined by using the method described in the previous study [66], and the A-Ci curves were measured at the following external CO_2_ concentrations in the order of 2000, 1500, 1000, 800, 600, 500, 400, 300, 200, 100, 50, and 0 μmol mol^−1^ with 300 μmol m^−2^ s^−1^ PAR. The light curves were measured at the photosynthetically active radiation (PAR) levels of 1500, 1200, 1000, 800, 600, 400, 200, 150, 100, 80, 50, 20, and 0 μmol m^−2^ s^−1^ with 450 ppm external CO_2_. The measurements were applied under the condition of leaf temperature at 21 °C, and relative humidity was ~50% in all measurements. 

### 4.7. Stomatal Density and Stomatal Complex Size Measurement

The stomatal density and stomatal complex size (including stomatal complex length and width, and stomatal pore length, width and area) of the abaxial epidermis of five-week-old plants were examined under a light microscope (TS100, Nikon, Tokyo, Japan) described in a previous study [67]. Stomata and pavement cell numbers were counted, and stomatal pore width and length, stomatal pore area, and stomatal complex length and width were measured with Image J 1.42 software (http://rsb.info.nih.gov/ij/, accessed on 3 March 2023). The abaxial epidermis of at least six plants was measured for each genotype per experiment. Experiments were repeated three times. For stomatal density and index measurements in cotyledons, the cotyledon epidermal layers from 15d after germination plants were stained with 10 µg/mL Propidium Iodide for 5min in darkness at room temperature, and then imaged using confocal microscopy (TCS-SP8; Leica, Weztlar, Germany) with 514 nm excitation and 590–630 nm emission.

### 4.8. Drought Stress Assays 

Drought stress assays were carried out using two-leaf stage *Bna.EPF2-OE* lines (each pot containing 16 plants with the same weight of soil and water content). Drought stresses were applied to the plants by withholding watering for 21 days until significant differences in the wilted leaves between different genotypes were observed, and then water was re-supplied for 5 days. Photographs of the plants at these three time points were taken. 

### 4.9. Biomass and Yield Trait Measurements

The WT and transgenic Brassica napus plants were cultivated in the experimental farm of the Huazhong Agriculture University, Wuhan, China, in 2021–2022. All the field experiments followed a randomized complete block design with three replications. Each genotype was cultivated to at least six lines. Sufficient irrigation was provided periodically, as needed to supplement rainfall. Routine management was carried out during the whole process of growth. Biomass and yield-related traits, including plant height, length of main inflorescence, number of siliques on the main inflorescence, number of first branch, length per silique, number of seeds per silique, 1000-seed weight, and seed yield per plant, were measured as described previously [68]. All data were collected at the mature stage in May 2022. All weight data were weighed with an electronic balance of ten thousandths, and all length data were recorded in centimeters using a measuring tape.

### 4.10. Statistical Analysis

All data were examined by Student’s *t*-tests. All values were presented as the means ± standard error of the mean (SEM) or standard deviation (SD). A *p*-value of <0.05 indicated a significant difference between means within WT and overexpression lines.

## Figures and Tables

**Figure 1 ijms-24-08007-f001:**
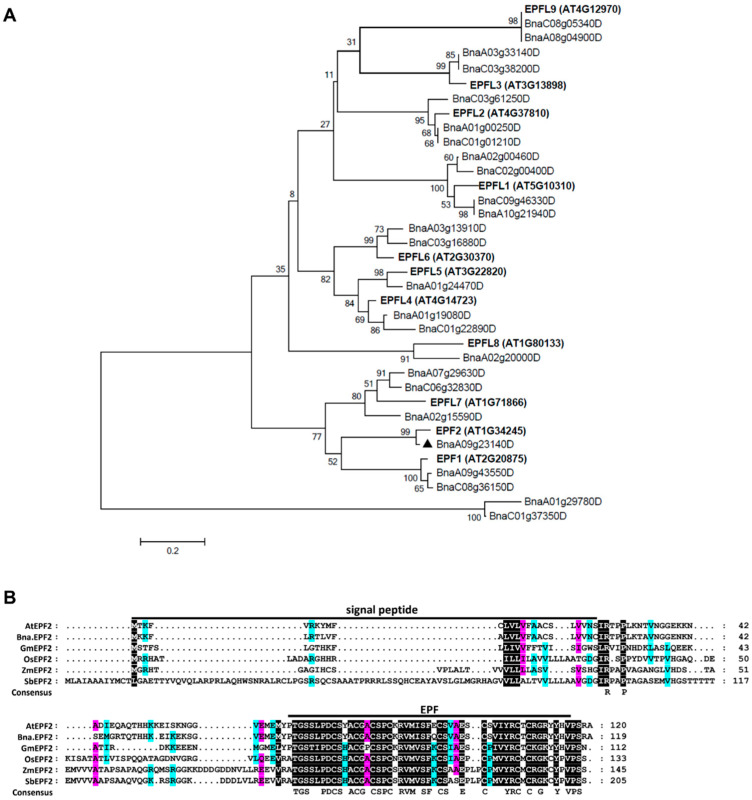
Phylogenetic tree of the EPF family in *Arabidopsis thaliana* and *Brassica napus,* and sequence alignment of EPF2 in different species. (**A**) The phylogenetic tree of the EPF family genes from *Arabidopsis thaliana* and *Brassica napus* were constructed using the MEGA6.06 software and maximum likelihood method. The black triangle represents *Bna.EPF2* in this study. (**B**) Amino acid sequence alignment of *EPF2* in *Sorghum bicolor* (Sb3006G233600), *Zea mays* (Zm00001eb431780), *Oryza sativa* (Os04g063730), *Brassica napus* (BnaA09g23140D), *Arabidopsis thaliana* (AT1G34245), and *Glycine max* (Gm08G168400). Alignment was performed with ClustalX (version 1.83) in multiple alignment mode and edited with GeneDoc 1.0 software (www.psc.edu/biomed/genedoc, accessed on 3 March 2023). Conserved regions were shaded in black (completely conserved), pink (very highly conserved), and blue-green (highly conserved) according to the conservation degree.

**Figure 2 ijms-24-08007-f002:**
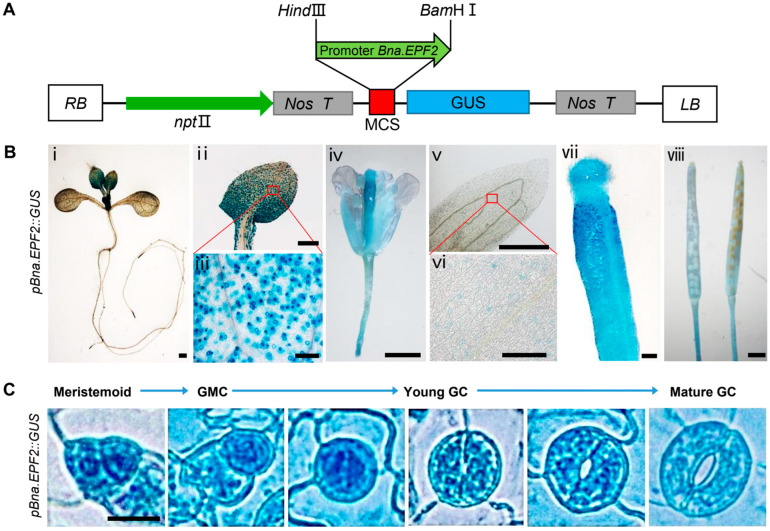
*Bna.EPF2* is highly expressed in stomatal lineage cells. (**A**) Construct for *GUS* reporter expression driving by the promoter of *Bna.EPF2.* (**B**) Histochemical GUS expression of *Bna.EPF2* in seedling (i), true leaves (ii,iii), flower (iv), calyx (v,vi), stigma (vii), and silique (viii). Bar = 0.5 mm in (i,ii,iv,v,vii,viii); Bar = 0.1 mm in (iii,vi); (**C**) GUS staining in stomatal lineage cells at different stages of stomatal development in *pBna.EPF2*:GUS transgenic plants. Bar = 10 µm. GMC, guard mother cell. GC, guard cell.

**Figure 3 ijms-24-08007-f003:**
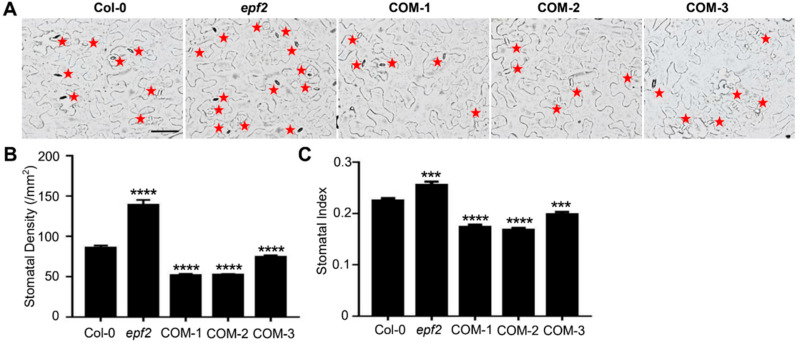
Ectopic expression of *Bna.EPF2* rescued the stomatal development defects in the Arabidopsis *epf2* mutant. (**A**) Imaging of stomatal density in col-0, Arabidopsis *epf2* mutant and *Bna.EPF2* complementation *epf2* lines, COM-1, COM-2, and COM-3. Red stars represent the stomata. Bar = 50 µm. (**B**,**C**) Statistical analyses of stomatal density (**B**) and stomatal index (**C**) in Col-0, *epf2*, and *Bna.EPF2* complemented *epf2* plants. Values are means ± SE (each with at least 20 leaves per experiment). ***, *p* < 0.001, ****, *p* < 0.0001, Student’s *t*-test.

**Figure 4 ijms-24-08007-f004:**
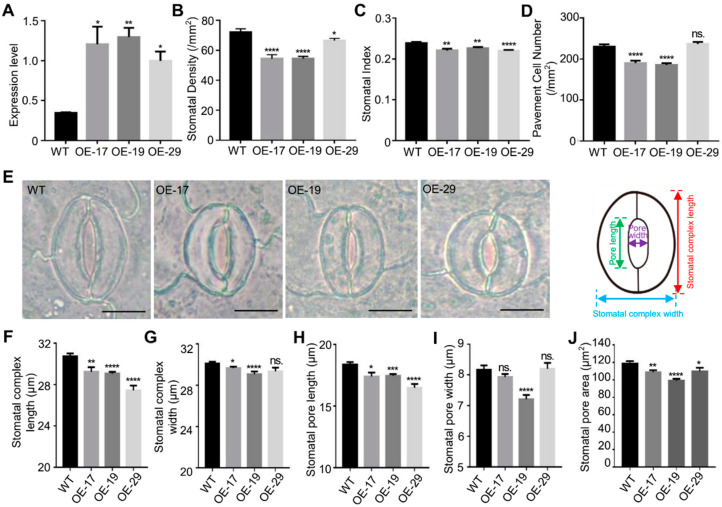
*Bna.EPF2* overexpression lines showed a reduced stomatal density, stomatal index, and smaller stomatal complex size compared to WT. (**A**) *Bna.EPF2* expression was measured by qPCR and normalized to *Actin* in Brassica. (**B**–**D**) Statistical analyses of stomatal density (**B**), stomatal index (**C**), and pavement cell density (**D**) in true leaves of five-week-old WT and *Bna.EPF2* overexpression lines. Values are means ± SE (each with at least 20 leaves per experiment). No significant difference (ns.), *p* ≥ 0.05; *, *p* < 0.05; **, *p* < 0.01; ****, *p* < 0.0001; Student’s *t*-test. (**E**) Imaging of stomatal complex in five-week-old WT and *Bna.EPF2* overexpression lines and model of stomatal complex. Bar = 10 µm. (**F**–**J**). Statistical analyses of stomatal complex size, such as stomatal complex length (**F**), stomatal complex width (**G**), stomatal pore length (**H**), stomatal pore width (**I**) and stomatal pore area (**J**) in five-week-old WT and *Bna.EPF2* overexpression lines. Values are means ± SE (each with at least 60 stomata analyzed per experiment). No significant difference (ns.), *p* ≥ 0.05; *, *p* < 0.05; **, *p* < 0.01; ***, *p* < 0.001; ****, *p* < 0.0001; Student’s *t*-test.

**Figure 5 ijms-24-08007-f005:**
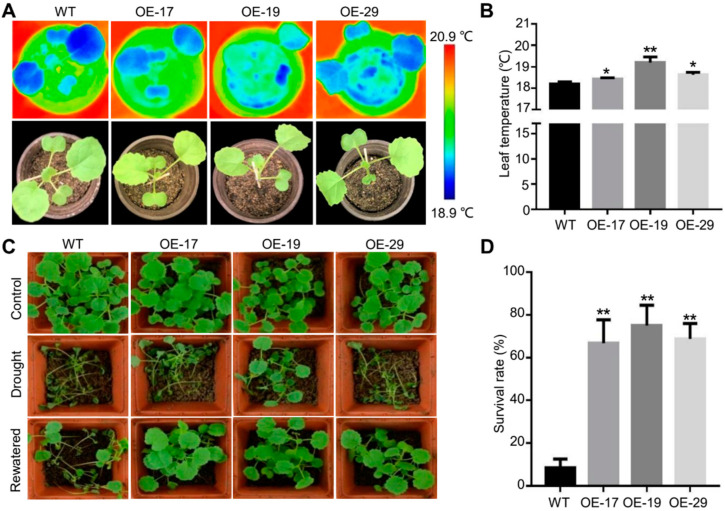
The *Bna.EPF2* overexpression plants showed a higher leaf temperature and stronger drought resistance. (**A**) Thermal imaging of fully expanded first and second true leaves of three-week-old Brassica plants under normal growth conditions using an infrared camera (FLIR systems; T420). The experiments were repeated three times. (**B**) Statistical analyses of leaf temperature of WT and *Bna.EPF2* overexpression plants using the software FLIR Tools+ ver. 5. 2. Values are means ± SE (*n* = 3). *, *p* < 0.05; **, *p* < 0.01; Student’s *t*-test. (**C**) Drought performance of two-leaf stage *Bna.EPF2* overexpression plants under drought stresses and water recovery conditions. Images were obtained before drought (control), withholding watering for 21 days (drought), and 5 days after re-watering (rewatered). The experiments were repeated three times. (**D**) Statistical analyses of the survival rate of WT and *Bna.EPF2* overexpression plants. Values are means ± SE (*n* = 3). *, *p* < 0.05; **, *p* < 0.01; Student’s *t*-test.

**Figure 6 ijms-24-08007-f006:**
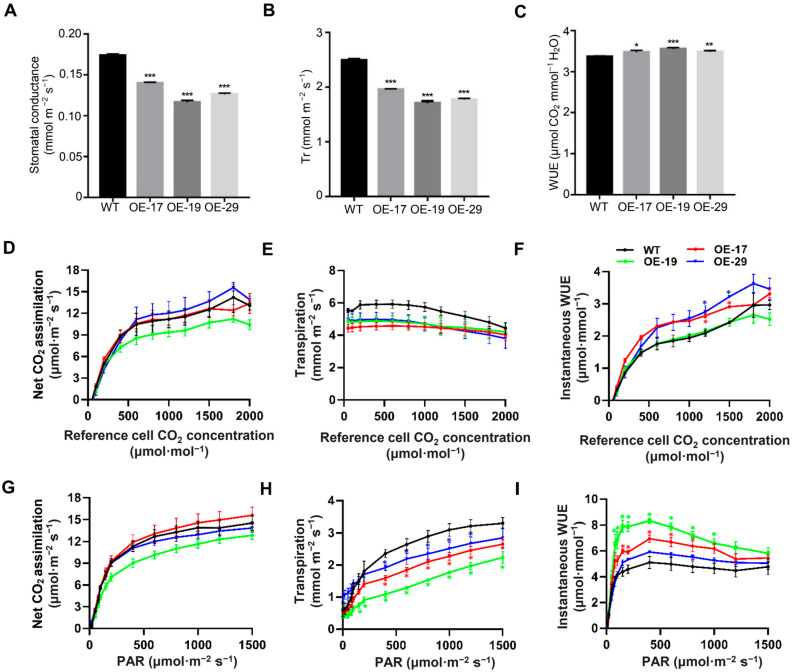
*Bna.EPF2* overexpression plants exhibited reduced stomatal conductance and transpiration under different conditions. (**A**–**C**) Stomatal conductance (mmol·m^−2^s^−1^) (**A**), leaf transpiration rate (Tr, mmol m^−2^s^−1^) (**B**), and instantaneous water use efficiency (iWUE, μmol CO_2_·mmol^−1^ H_2_O) (**C**) in five-week-old well-watered *Bna.EPF2* overexpression lines. (**D**–**F**) Net CO_2_ assimilation (**D**), transpiration (**E**), and instantaneous WUE (**F**) in five-week-old well-watered *Bna.EPF2* over-expression lines in response to shifts in CO_2_ concentrations. (**G**–**I**) Net CO_2_ assimilation (**G**), transpiration (**H**), and instantaneous WUE (**I**) in five-week-old well-watered *Bna.EPF2* over-expression lines in response to shifts in light intensity. All data were measured by a Li-6400XT gas exchange system. *n* ≥ 3 leaves per genotype per experiment, experiments were repeated three times, values are means ± SE, *, *p* < 0.05; **, *p* < 0.01; ***, *p* < 0.001; Student’s *t*-test, asterisk (*) with different colors present the significant difference of the different *Bna.EPF2* overexpression lines compared to WT at *p* < 0.05 level.

**Table 1 ijms-24-08007-t001:** The effect of *BnA.EPF2* overexpression on Brassica biomass and yield traits.

	WT (*n* = 6)	*Bna.EPF-17* (*n* = 6)	*BnaEPF-19* (*n* = 6)	*BnaEPF-29* (*n* = 6)
Plant height (cm)	140.67 ± 8.08 a	142.17 ± 9.87 a	132.67 ± 8.24 a	147.75 ± 9.01 a
		(*p* = 0.5737)	(*p* = 0.1138)	(*p* = 0.1919)
Length of main inflorescence (cm)	42.33 ± 6.53 a	39.92 ± 5.50 a	36.67 ± 6.62 a	41.33 ± 6.28 a
		(*p* = 0.5039)	(*p* = 0.1665)	(*p* = 0.7924)
Number of siliques on main inflorescence	39.50 ± 4.76 a	37.83 ± 4.88 a	37.17 ± 2.48 a	38.67 ± 5.13 a
		(*p* = 0.5626)	(*p* = 0.3124)	(*p* = 0.7765)
Number of first branch	9.00 ± 1.79 a	8.83 ± 1.17 a	8.67 ± 1.75 a	8.33 ± 2.25 a
		(*p* = 0.8523)	(*p* = 0.7510)	(*p* = 0.5826)
Length per silique (cm)	5.16 ± 0.33 a	5.38 ± 0.28 bc	5.31 ± 0.25 b	5.47 ± 0.31 c
		(*p* = 0.0001)	(*p* = 0.0049)	(*p* = 0.00008)
Number of seeds per silique	24.45 ± 2.79 a	25.42 ± 2.47 b	24.73 ± 2.48 ab	26.15 ± 3.00 b
		(*p* = 0.0468)	(*p* = 0.5573)	(*p* = 0.0017)
1000-seed weight (g)	3.49 ± 0.34 a	3.28 ± 0.44 ab	3.27 ± 0.32 ab	3.66 ± 0.45 b
		(*p* = 0.3952)	(*p* = 0.2797)	(*p* = 0.0427)
Seed yield per plant (g)	20.92 ± 1.16 a	20.52 ± 1.60 a	20.36 ± 1.77 a	20.60 ± 1.48 a
		(*p* = 0.3431)	(*p* = 0.1748)	(*p* = 0.1138)

The values represent the means of six replicates. Different letters next to the number indicate significant difference at *p* ≤ 0.05 by Student’s *t*-test.

## Data Availability

No large datasets were created in this study.

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
