# Peer review of "Bna.EPF2 Enhances Drought Tolerance by Regulating Stomatal Development and Stomatal Size in Brassica napus"

_ijms, 2023, doi:10.3390/ijms24098007_

Round 1

Reviewer 1 Report

The topic of this work is important. The experiments are well designed and executed, and the conclusions are well supported by the data provided.

Reviewer 2 Report

I have read the manuscript (ijms-2292434). Titled: Bna.EPF2 Enhances Drought Tolerance by Regulating Stomatal Development and Stomatal Size in Brassica napus for publication in International Journal of Molecular Sciences MDPI. Generally, the authors present an interesting study. The paper is well written and presented. However, there are some problems in this study. Therefore could be acceptable after major revision for possible publication in this journal, but, some corrections should be considered by authors. I request this manuscript should be correct by author in this stage. Thank you

Reviewer 3 Report

Jiao and colleagues preliminarily assessed the role of one EPF gene in regulating the B. napus response to drought stress. It makes sense for a more extensive understanding of the molecular mechanisms under the plant stress response. The experiments were overall well designed and performed, and results were presented and discussed properly. However, due to some questions remaining, I am sorry to say this manuscript is not acceptable for publication in its current state.

1.      The language of this manuscript should be improved. Gramma mistakes could be found through. For example, line 46, ‘generate’ should be ‘generates’; line 48 whereas Mshould be “whereas an M”; line 109, “is a most closed” should be “is the most closed”; line 159-161 & line207-209, these sentences should be rephrased. Line535, “Zea_mays” and oryza_sativashould be “Zea mays” and Oryza sativa.

2.      Line 68, “… tomato”, the related references should have been cited.

3.      The authors stated that they used TBLASTN to search the Arabidopsis homologs in B. napus. I suppose they want to find any locus missed during genome annotation. It’s all right,but please describe the searching strategy in detail in the section methods, as after a repeat by myself, I do find at least two copies of the EPF2 in A9 and C6, in contrast to the description in this manuscript that only one copy was found.

4.      Some figure legends are too wordy, such as that for Fig 6.

5.      Line 112, a more detailed description of the cleavage site prediction is needed, if the authors think this information is of some value.

Reviewer 4 Report

The methods description must be improved, and specify the possible variations, done or identified as necessaries, from the reference cited.

The approach for overexpression in poorly explained; the only mention of the vector used is its cite, only one time in all document. The same situation occurs in description of promoters etc. Which is coherent with the general style from all paper but limits the understanding from readers less familiarized specifically with genetic plant enhancement/ transformation. This situation could be improved with a little more extend "medium-high level" explanation in the corresponding sections.

Improve the analysis respect to the impact of agronomic traits of interest, and not only respect to “seed yield per plant”. For example, by mean of the comparative analysis for, all or some from, the following parameters:

·      chlorophyll contents

·      fatty acids composition or at least iodine value

·      affection to glucosinolates content, especially relevant for B.napus

·      oil content

·      Protein/ nitrogen content would equally be informative for the repercussion on plants/seed for others possible uses different to oil production.

Applied utility of the paper apart, if well these complementary analyses could be considered not necessaries according to the “main” conclusions/ “revindications” from paper (seed yield per plant), in other parts of the document this point is referred in a wider as “yield production/ yield traits” (from abstract). This should be revised to avoid conflicts respect to supporting of conclusions from results.

The insertion of figures and graphs in their corresponding position (“in line”) according to the text contents would make the work easier to read and analyze.

Some arbitrary changes of font size and in the use of upper and lowercase letters.

Reviewer 5 Report

Some parts of the paper should be rewritten in a different way in order to be more clear.   The aim of the study must be included in the last paragraph of the introduction, so this last paragraph must be rewritten. In the discussion , the results of relevant studies should be included , so the discussion needs a revision

Round 2

Reviewer 2 Report

Now the authors have satisfactorily addressed all the concerns and revised the manuscript accordingly. The manuscript could be accepted for its publication in International Journal of Molecular Sciences.